# Design of Trials for Cerebral Small Vessel Disease and Vascular Cognitive Impairment

**DOI:** 10.3390/neurolint17110181

**Published:** 2025-11-04

**Authors:** Elizabeth Phan, Shi Pei Loo, Terence J. Quinn

**Affiliations:** Academic Section of Geriatric Medicine, University of Glasgow, Glasgow Royal Infirmary, Glasgow G4 0SF, UK; shipei.loo1@gmail.com (S.P.L.); terry.quinn@glasgow.ac.uk (T.J.Q.)

**Keywords:** cerebral small vessel disease, vascular cognitive impairment, vascular dementia, clinical trials, trial design

## Abstract

**Background/Objectives:** Cerebral small vessel disease (cSVD) and vascular cognitive impairment (VCI) are major contributors to stroke and dementia. Despite their importance, there are few effective treatments for cSVD and VCI. Variability in cSVD/VCI populations, intervention targets, and outcome selection may contribute to inconsistencies and challenges in clinical trial design. We reviewed the design of cSVD and VCI clinical trials to describe current practice in the selection of populations, interventions, and outcomes. **Methods:** We systematically searched Ovid Medline, Embase, and PsychInfo databases for recently completed cSVD/VCI trials and searched online trial registries (ClinicalTrials.gov, European Union Clinical Trials Register, and International Clinical Trials Registry Platform) for ongoing cSVD/VCI trials. We determined the use of specific categories of inclusion and exclusion criteria, interventions, and outcomes in the included trials and described these as counts and percentages. **Results:** We included a total of 82 cSVD trials and 120 VCI trials. Neuroimaging features were most frequently used as inclusion criteria for cSVD (88%) and cognition for VCI (88%). There was substantial variation in eligible ages for participation. Both cSVD and VCI trials largely excluded patients with comorbidities, vascular risk factors, and neuropsychiatric disorders, with a notable proportion of cSVD trials excluding on the basis of functional impairment. The most studied intervention classes were repurposed cardiovascular drugs (40%) for cSVD and Traditional Chinese Medicine (35%) in VCI. The most common primary outcome category was neuroimaging for cSVD (53%) and cognition for VCI (86%). Notably, functional outcomes were underused in both cSVD and VCI trials (13% and 12%, respectively, for primary outcomes). **Conclusions:** We have identified substantial variability in all aspects of cSVD and VCI clinical trial design. Inconsistent neuroimaging criteria and exclusions based on common long-term conditions limit the generalisability of findings. There is a need for greater focus on clinical outcomes, particularly functional ability, to better reflect treatment impact. Increased integration and standardisation of cSVD and VCI trial design is needed to accelerate progress in developing treatments.

## 1. Introduction

Cerebral small vessel disease (cSVD) and vascular cognitive impairment (VCI) are important and related conditions. cSVD is a chronic and cumulative brain disease affecting the smallest cerebral blood vessels [1]. cSVD is usually recognised as a combination of neuroimaging features, including small subcortical infarcts, white matter hyperintensities, lacunes, microbleeds, and enlarged perivascular spaces [2]. cSVD may present as chronic functional and cognitive decline associated with vascular risk factors and/or repeated acute lacunar strokes. VCI refers to loss of cognitive function caused by cerebrovascular insults. The severity of VCI ranges from mild cognitive impairment (MCI) to vascular dementia (VaD). VCI and cSVD frequently co-exist. Indeed, cSVD is the most common cause of VCI [3].

cSVD is a highly heterogeneous disease both in terms of its pathology and its presentation. Sporadic cSVD is classified into cerebral amyloid angiopathy (CAA) and non-amyloidal sporadic cSVD, the latter being more common in the ageing population. There are also genetic forms of cSVD, such as the monogenic disorder cerebral autosomal dominant arteriopathy with subcortical infarcts and leukoencephalopathy (CADASIL). Clinically, cSVD may include covert cSVD (asymptomatic neuroimaging features), acute lacunar stroke syndromes, or chronic symptoms such as cognitive impairment, gait disturbances, mood disorders, and vascular parkinsonism [4]. Despite attempts to standardise radiological terminology in cSVD, for example, through The Standards for Reporting Vascular changes on Neuroimaging (STRIVE) guidance documents [5,6], clear diagnostic criteria distinguishing cSVD clinical phenotypes remain elusive.

The VCI label also encompasses highly heterogeneous populations. VCI represents a continuum of cognitive impairment, from vascular mild cognitive impairment to vascular dementia (VaD), and does not differentiate patients with other clinical features such as neuropsychiatric symptoms and functional impairment. Several diagnostic criteria exist, including the National Institute of Neurological Disorders and Stroke-Association Internationale pour la Recherché et I’Enseignement en Neurosciences (NINS-AIREN) criteria, the State of California Alzheimer’s Disease Diagnostic and Treatment Centres (ADDTC) criteria, the DSM-5 criteria, and the ICD-10 criteria. These criteria have been criticised for their over-emphasis on memory impairment and variability in requirements for severity of cognitive impairment, classification of strokes, and neuroimaging features [7]. This consequently translates into significant variability of clinical trial populations.

Many hypotheses suggest that cSVD pathogenesis involves a combination of factors, including blood–brain barrier disruption, endothelial dysfunction, and inflammation. Such processes lead to microhaemorrhages and ischaemia, causing neuronal death [8]. Similar mechanisms are also likely to contribute to the neurodegeneration seen in other forms of VCI. Other pathophysiological processes leading to VCI include atherosclerotic changes, systemic metabolic disturbances such as dysglycaemia, and deposition of aberrant proteins seen in Alzheimer’s disease (AD) (Amyloid β and tau) [9]. These underlying mechanisms of pathology present several possible treatment targets, including cardiovascular disease preventative agents, anti-inflammatories, and endothelial stabilising drugs. However, many treatments with plausible underlying science have failed to show efficacy in randomised controlled trials (RCTs) in cSVD and VCI. This may be due to the interventional agent, the intrinsic heterogeneity of the populations, or problems with clinical trial designs.

Variation in trial design has been demonstrated in both cSVD and VCI RCTs. The Framework for Clinical Trials in Cerebral Small Vessel Disease (FINESSE) study identified several aspects of heterogeneity in current trials. These included poorly defined populations recruited from various clinical specialties, inconsistent selection criteria, and unsuitable outcome measures [10]. Smith et al. (2017) conducted a systematic review of VCI clinical trials, which identified heterogeneous patient populations relating to diagnostic criteria and disease severity, lack of prespecified primary outcomes, and variability in types of cognitive assessment tools [11].

There is a need to identify suitable outcomes that meaningfully demonstrate patient-centred treatment efficacy, while considering costs, feasibility of data collection, and interpretability of results [10]. Expert panel consensuses, such as the European Stroke Organisation (ESO), have recommended the use of clinical endpoints (such as functional and cognitive assessments) over surrogates (such as neuroimaging and biomarkers) in cSVD trials [5]. However, heterogeneity within cSVD/VCI populations may complicate the prioritisation of specific outcomes. For example, functional assessments may be less relevant in populations with cSVD-related neuroimaging features but no overt clinical symptoms, while imaging-based outcomes may hold less significance for patients with severe impairments and quality-of-life burden. Even with prioritisation of clinical endpoints, there is also inter-outcome variability in terms of types of clinical assessments used. A 2023 systematic review on 118 VCI clinical trials identified 125 different outcome measures used across neuropsychological, functional, and instrumental categories [12]. This inconsistency in outcome measures limits meaningful comparison between trials to determine treatment efficacy and future trial designs [13].

Given this increasing availability of guidance around research design in the cSVD/VCI space, and acknowledging the dynamic nature of clinical trials, a contemporary appraisal is needed. A first step towards creating consensus and raising standards is to describe current practice in clinical trials with a cSVD/VCI focus. In trial design, evidence synthesis, and guidelines, the PICO (Population, Intervention, Control, and Outcome) framework is often used to describe trials. Thus, our specific areas of interest were included (and excluded) populations, the interventions studied, and the methods used to assess treatment effects.

In this study, we reviewed the methods of non-amyloidal sporadic cSVD and VCI trials to describe populations, interventions, and outcomes.

## 2. Materials and Methods

Scoping reviews were conducted to collate information on the methods used in completed and ongoing trials in cSVD and VCI. Our reporting is guided by the Preferred Reporting Items for Systematic Reviews and Meta-Analyses extension for Scoping Reviews (PRISMA-ScR) checklist, adapted to account for the structure of this project as a methodological scoping review [14]. Certain items of the PRISMA-ScR checklist were not applicable (e.g., critical appraisal) as the aim of this study was to conduct a descriptive synthesis of populations, interventions, and outcomes used in cSVD and VCI trials, rather than evaluate the quality of studies and efficacy of interventions. The review protocol was registered on the International Platform of Registered Systematic Review and Meta-analysis Protocols resource (INPLASY—Registration Number: INPLASY202590013, doi:10.37766/inplasy2025.9.0013).

### 2.1. Eligibility Criteria

#### 2.1.1. Types of Studies

Both randomied and non-randomised clinical trials were included, provided they were studies of an intervention. To offer a representation of contemporary trial activity, we included both recently completed and ongoing studies. Completed trials with published full-text papers were included based on literature searching. For ongoing trials, trials at any stage of recruitment status, or complete trials with unpublished results, were included. Non-interventional designs, such as observational studies, systematic reviews, and test accuracy studies, were excluded. Conference abstracts were excluded. Secondary and subgroup analyses of larger trials were excluded if the primary focus of the parent trial was not cSVD/VCI. Duplicate titles were excluded; if a trial was described in more than one paper, the paper with the most detailed description was included. No language restrictions were enforced, but foreign language papers where we did not have access to a suitable translator or equivalent English text were not included in the final analysis.

#### 2.1.2. Types of Participants

Inclusion and exclusion criteria were adjusted to the two populations of interest. For cSVD, trials with adult human participants of any age, with any non-amyloidal sporadic cSVD presentations, were potentially included, including lacunar stroke and any of the neuroimaging features described in STRIVE guidance. Trials with mixed or non-lacunar pathologies, other neurodegenerative diseases, and genetic or amyloidal cSVD types were excluded. These rarer forms of cSVD [15] were excluded as they have unique pathophysiological mechanisms that may necessitate differing trial designs compared to the more common sporadic cSVD.

For VCI, trials with adult human participants of any age with VCI, including vascular dementia (VaD), were potentially included, including post-stroke cognitive impairment and cognitive impairment caused by cSVD. Other types of dementia, such as Alzheimer’s and frontotemporal dementia, were excluded. Trials with ‘mixed dementia’, or mixed populations with different dementias, were also excluded.

#### 2.1.3. Types of Interventions and Outcomes

Trials studying any pharmacological and/or non-pharmacological interventions to treat or prevent non-amyloidal sporadic cSVD, or VCI, were potentially included. Trials could be at any stage of development from Phase I (first-in-man) through to Phase IV (post-licence surveillance). No restrictions were imposed on outcomes.

### 2.2. Search Strategy and Study Selection

Two complementary searches were performed for cSVD and VCI. Updates were performed for both searches, and the last full search was conducted in August 2023. Multidisciplinary databases were searched, from 2012 to the last search: MEDLINE (Ovid), EMBASE (Ovid), PsycInfo (EBSCO).

Clinical trial search filters and topic syntax developed by the Scottish Intercollegiate Guidelines Network (SIGN) and Cochrane were used. As there were no standardised, consensus search terms for cSVD, a search syntax was developed by adapting suitable terms from the ‘keywords’ and ‘search strategy’ sections of cSVD reviews, found through initial literature scoping [1,2,4,5,6,8,9]. The year 2012 was chosen as the inception point, to represent contemporary trials and due to the term ‘cSVD’ being characterised in 2009 [6].

Ongoing trials were searched for using the following online trial registries: ClinicalTrials.gov, the International Clinical Trials Registry Platform (ICTRP), and the European Union Clinical Trials Register (EUCTR). The advanced search tool was used where possible. The main disease (if applicable) in the search was ‘cerebral small vessel disease’, with other terms from our cSVD syntax used as well, until no new studies were found. Searches for trials using the terms ‘vascular cognitive impairment’ and ‘vascular dementia’ were conducted separately.

Eligibility assessment for the cSVD and VCI trial searches was conducted by two reviewers working independently to screen titles, abstracts, and study protocols. The project’s principal investigator was consulted to resolve any uncertainty regarding eligibility.

Details on search terms are provided in the Appendix A.

### 2.3. Data Extraction and Synthesis

Data from included trials were extracted into Microsoft Excel 2020 files for cSVD and VCI trials. For each file, five data extraction sheets were created. One sheet contained basic trial information, including registry identifiers, (expected) date of completion, number of participants, interventions, comparators, trial status, and phase. Due to the variability of cSVD and VCI populations in trials, we created a categorization system describing the cSVD and VCI subtypes encountered, shown in Table 1. The other four sheets contained data relating to selection criteria (type of patient population), interventions, and outcomes.

Trials were assessed on whether they used specific categories of inclusion criteria, exclusion criteria, primary and secondary outcomes, and interventions. These categories were developed through an initial assessment of the methodologies of included trials. Final categories for each data extraction sheet are as follows:

Inclusion/Exclusion Criteria: Comorbidities, cognition, functional status, neuroimaging, neurological impairments, neuropsychiatric and mood disorders, physiological measurements, stroke occurrence (clinical evidence or previous history of any stroke type before enrolment into trial), tissue biomarkers, vascular risk factors, Hachinski Ischaemic Scale (VCI only), and other selection criteria.

Interventions: Antihypertensives, antithrombotics, cognitive enhancers/anti-dementia drugs, vasoactive drugs, cognitive rehabilitation, physiological interventions, exercise and lifestyle interventions, Traditional Chinese Medicine, other pharmacological interventions, and other non-pharmacological interventions.

Outcomes: Clinician assessment, cognition, economic outcomes, functional outcomes, neuroimaging (other), neuroimaging (structural), neurological impairments, neuropsychiatric and mood disorders, physiological measurements, safety and adherence outcomes, stroke occurrence (occurrence of a clinical stroke event from the start of the trial), tissue biomarkers, and other outcomes.

Definitions of categories are provided in the data dictionary in the Appendix A.

Additionally, for trials that used cognitive and functional outcomes as primary outcomes, we tabulated the frequency of different types of cognitive and functional assessment tools used in these trials. This was to demonstrate patterns of heterogeneity not only in outcome categories used but also in specific outcome measures. In categorising the cognitive outcomes, we recognised three differing assessment approaches—multi-domain screening tools, tools designed to assess a single cognitive domain, and detailed multi-domain assessments (referred to as neuropsychological batteries).

### 2.4. Data Presentation

All basic characteristics of trials, and categories used as selection criteria, interventions, and outcomes were summarised descriptively using counts and percentages. The Microsoft Excel files containing the raw data are available in the Appendix A.

## 3. Results

### 3.1. Study Selection

#### 3.1.1. cSVD Study Selection

The initial search for completed cSVD trials yielded 640 titles, of which 573 titles were excluded, leaving 67 titles for abstract screening. These were reduced to 19 completed trials. 9 completed trials were added from the search of trial registry databases, totalling 28 completed trials for review.

For ongoing cSVD trials, the initial search yielded 35 trials from ClinicalTrials.gov, 12 from EUCTR, and 175 from ICTRP (the ICTRP advanced search tool could not narrow searches to clinical trials only). After title screening, 1 trial from ClinicalTrials.gov, 9 from EUCTR, and 81 from ICTRP were discarded. After protocol screening, 80 trials remained, and after review, 54 ongoing cSVD trials were included. Thus, 82 cSVD studies in total were reviewed.

Study selection processes for completed and ongoing cSVD trials are summarised in Figure 1.

#### 3.1.2. VCI Study Selection

The initial search for completed VCI trials yielded 1026 titles, of which 683 titles were excluded, leaving 343 abstracts for screening. After exclusion of 291 abstracts, 56 full-text papers were screened. A total of 38 completed trials were included.

From the trial registries, there was an initial yield of 562 trial protocols, from which 480 were excluded, leaving 82 ongoing trials to be included. Thus, 120 VCI studies in total were reviewed.

Study selection processes for completed and ongoing VCI trials are summarised in Figure 2.

### 3.2. Basic Characteristics

#### 3.2.1. cSVD Trials

Basic characteristics of the cSVD trials are summarised in Table 2. Of trials with a reported phase, most were in Phase 4. Most trials were conducted in Asia. The most represented cSVD subtypes were symptomatic neuroimaging abnormalities and lacunar stroke.

#### 3.2.2. VCI Trials

Basic characteristics of VCI trials are summarised in Table 3. There were relatively similar proportions for Phases 2 to 4. In absolute numbers, the most represented subtype was undifferentiated VaD, but there was a relatively similar distribution of VCI subtypes (stroke-associated, cSVD-associated, and undifferentiated).

### 3.3. Patient Populations

#### 3.3.1. cSVD Patient Population

Table 4 summarises the counts and percentages of cSVD trials according to inclusion and exclusion criteria. Neuroimaging features were most frequently used as inclusion criteria, followed by cognition, stroke occurrence, neurological impairments, and functional status. Apart from co-morbidities, vascular risk factors were most frequently used as exclusion criteria in cSVD trials, although a small but notable percentage of trials also used vascular risk factors as inclusion criteria. Large percentages of cSVD trials also used cognition, stroke occurrence, and neuropsychiatric and mood disorders as exclusion criteria.

#### 3.3.2. VCI Patient Population

Table 5 summarises the counts and percentages of VCI trials according to inclusion and exclusion criteria. Cognition was most frequently used as inclusion criteria, followed by neuroimaging features, stroke occurrence, the Hachinski Ischaemic Scale, vascular risk factors, and functional status. Apart from co-morbidities, neuropsychiatric and mood disorders were most frequently used as exclusion criteria, followed by cognition, vascular risk factors, and neurological impairments.

### 3.4. Interventions

Table 6 summarises counts and percentages of cSVD and VCI trials using each intervention class. The most represented intervention class was physiological interventions (examples include physiotherapy and ischaemic conditioning) in cSVD trials and Traditional Chinese Medicine (TCM) in VCI trials. Despite their frequent study in stroke trials, antihypertensives, antithrombotics, and vasoactive drugs comprised only a small proportion of the interventions tested.

### 3.5. Outcomes

#### 3.5.1. cSVD Outcomes

Table 7 summarises counts and percentages of cSVD trials using specific outcome categories. The most frequently used primary outcome category for cSVD trials was cognition, followed by structural neuroimaging features, non-structural or other neuroimaging features, and physiological measurements. Notably, although functional outcomes are mandated for stroke trials, only a modest proportion of cSVD trials used functional outcomes as a primary outcome. There was a wider distribution of outcome categories used as secondary outcomes. The most frequently used secondary outcome category was also cognition, followed by physiological measurements, and safety and adherence outcomes.

Table 8 and Table 9 summarise cognitive and functional assessment tools used as primary outcomes in cSVD trials. The Montreal Cognitive Assessment (MoCA) was the most frequently used cognitive assessment. The majority of trials using a functional assessment as a primary outcome did not specify the functional assessment tool used.

#### 3.5.2. VCI Outcomes

Table 10 summarises counts and percentages of VCI trials using specific outcome categories. The majority of VCI trials used cognition as primary and/or secondary outcomes. The other primary outcome categories were represented in less than 20% of the total VCI trials, and functional outcomes were infrequently a primary outcome. The most represented secondary outcome category was also cognition, followed by functional outcomes and neuropsychiatric disorders.

Table 11 and Table 12 summarise cognitive and functional assessment tools used as primary outcomes in VCI trials. Out of the assessment tools that were specified, we identified 46 different cognitive tools and 8 different functional tools. The most frequently used cognitive assessment tools were MoCA and the Mini-Mental State Examination (MMSE).

## 4. Discussion

The review of contemporary trials in cSVD and VCI demonstrates certain common features around populations, interventions, and outcomes, but also highlights substantial heterogeneity. The differences in trials for cSVD and VCI might suggest that their respective clinical and research communities view them as separate diseases. However, there is significant overlap in their pathogenesis and sequelae. This is leading to increasing interest in redefining VCI as a spectrum of ‘vascular cognitive disorders’ [16], ranging from mild cognitive impairment to vascular dementia, with cSVD being recognised as a predominant cause of the disease.

### 4.1. Patient Populations

Neuroimaging was widely used as an inclusion criterion, perhaps unsurprising given that neuroimaging is required for many cSVD and VCI diagnostic and classification criteria [17]. The FINESSE statements indicated 100% expert agreement that neuroimaging should be used to confirm cSVD [10]. However, even with the rubric of neuroimaging, there was substantial variability in definitions and features of interest. The STRIVE guidelines were first published in 2013. Nonetheless, several trials did not mention STRIVE or used an alternative, sometimes bespoke set of neuroimaging criteria.

There was substantial variation in the ages of participants eligible for studies. This may represent that trials in cSVD and VCI may have differing purposes and hence populations of interest. For example, a trial of prevention will likely require a younger age group [18] than a trial of treatment of an established disease. While lower age limits have a biological rationale, as paediatric pathologies and responses to treatment are different from adults, there is less of a compelling argument to exclude on the basis of an upper age limit alone. Frailty and functional decline may be reasonable reasons to not consider participation in a trial, but these are not synonymous with chronological age. Eliminating restrictive age limits would enable trials to observe the effect of interventions across the disease spectrum and in a more representative population, since both sporadic cSVD and VCI are age-associated conditions [19]. The exclusion of potential participants on the basis of age alone has been criticised by many professional organisations as representing a form of ageism, and best practice guidance is available to increase the inclusion of older adults in clinical research [20].

In designing any trial, there is tension between studying a scientifically ‘pure’ population and looking at effects in populations that are closer to the real world of clinical practice. The need for one approach or another may partly depend on the phase of the trial, with first-in-man studies likely to favour a more select population. It is notable that many studies were excluded on the basis of vascular or other comorbidities. However, many living with cSVD and/or VCI are multimorbid [21]. Where a trial intervention may have direct or indirect effects on comorbidity, for example, trials of cardiovascular secondary prevention, such exclusions may result in misleading estimates of the overall effect. While absolute rates of outcomes such as mortality or adverse events may be higher in those with multiple long-term conditions, there is limited evidence that comorbidities directly modify the treatment effects of most interventions. Exclusion on the basis of functional ability was another relatively common criterion. Trials rarely give the rationale for such exclusions, and one would hope that exclusion had a scientific rationale; for example, in trials of prevention, those with functional limitation, and hence more advanced disease, could be reasonably excluded, rather than using these exclusions to make recruitment and follow-up easier for trial teams, such as excluding those who may struggle to attend a remote test centre.

### 4.2. Interventions

There was notable variability in the interventions used in both cSVD and VCI trials. The majority of cSVD trials used repurposed cardiovascular drugs (antihypertensives, antithrombotics, and vasoactive drugs). Interest in cardiovascular drugs for cSVD is understandable; repurposed drugs are cheaper, and regulatory approvals for licencing are less burdensome as the medications have already been tested for safety. In contrast, cardiovascular drugs were less frequently used in VCI trials, with there being a greater emphasis on cognitive interventions. This may represent differing levels of equipoise, and many may think that the case for cardiovascular medications in VCI is already made. However, while several cardiovascular drugs have also shown potential neuroprotective effects, such as reducing white matter hyperintensities (WMHs) and reducing blood–brain barrier (BBB) damage [22], the clinical efficacy of cardiovascular drugs for neurocognitive disease remains unproven. Indeed, the European Stroke Organisation (ESO) guidelines for covert cSVD (ccSVD) recommend against the use of antihypertensives in normotensive ccSVD patients and antiplatelets in ccSVD patients with no other indications [5]. Conversely, vasoactive agents such as nitric oxide donors and phosphodiesterase inhibitors, especially when used in combination, have been shown to improve cognition and functional ability as demonstrated by the LACI-2 trial [23].

Physiological interventions (referring to treatments that alter physiology, such as ischaemic conditioning and physiotherapy) were used more frequently in cSVD trials than VCI trials, while exercise and lifestyle interventions (referring to programmes that alter patients’ lifestyles through dietary changes, stress management, and physical activity) were used more frequently in VCI trials. The popularity of physiological interventions may be attributed to the view of cSVD as a stroke syndrome, leading to increased interest in secondary prevention interventions (such as ischaemic conditioning) and stroke rehabilitation. The infrequency of exercise and lifestyle interventions in cSVD trials is, however, surprising. In addition to improving cardiovascular health, regular exercise is associated with markers of brain health and milder cSVD; examples of positive attributes are greater brain volume, improved cerebral blood flow, and prevention of endothelial dysfunction [24,25]. Increasing physical activity has been shown to be more effective for the prevention of Alzheimer’s disease dementia than altering other modifiable risk factors, including smoking, obesity, depression, diabetes, and hypertension [25,26]. Given the low cost and safety of exercise and lifestyle interventions, it seems surprising that there are so few trials in cSVD.

Traditional Chinese Medicine (TCM) was the most frequently tested intervention in VCI trials. TCM has a long history of development, and its use in China is widespread, with increasing interest internationally in recent years. However, robust evidence on its efficacy in VCI is limited [27]. TCM theory is underpinned by a holistic approach to treating and preventing disease, with multi-ingredient agents having various mechanisms of action [28]. Proposed mechanisms in VCI include modulating cholinergic content, reducing glutamate toxicity, decreasing neuroinflammation, preserving the blood–brain barrier damage, and inhibiting neuronal apoptosis. TCM also presents a favourable option due to its low cost and good safety profile [29]. A Cochrane review reported that two TCM formulations significantly improved Mini-Mental State Examination (MMSE) scores in vascular dementia compared to conventional drugs. However, the review concluded that the overall evidence in support of TCM use in VaD was still moderate to very weak due to a lack of high-quality, large-scale RCTs [27]. Furthermore, there is considerable variability in TCM formulations and dosing, leading to difficulty forming evidence synthesis through meta-analyses [29]. Hence, this further highlights the need for more standardised high-quality RCTs with robust reporting practices to fully establish TCM as an effective intervention for VCI. There are recent examples of high-quality, well-reported TCM trials in cerebrovascular medicine, albeit the results of intervention efficacy have been neutral [30].

### 4.3. Outcomes

Both cSVD and VCI tended to focus more on surrogates than on clinical endpoints. The most frequently used primary outcome in cSVD trials was neuroimaging. This included a large proportion of trials that used advanced neuroimaging modalities not commonly employed in clinical practice, limiting global scalability. Beyond cognitive assessment, other clinical outcomes, such as functional assessment, neurological impairments, and incident neuropsychiatric disorders, were infrequently used as primary outcomes, despite the importance ascribed to such outcomes by those living with the diseases. Notably, even amongst trials that used clinical endpoints (cognitive and functional outcomes) as primary outcomes, there appeared to be substantial variability in the types of outcome measures used.

Surrogate endpoints are useful as they are potentially more sensitive to change, allowing for smaller sample sizes, and they can offer clues as to underlying disease mechanisms. This may be a necessity in Phase 2 and earlier trials. However, in many fields, including stroke, clinical outcomes are preferred for Phase 3 and beyond [10,31]. Notably, the ESO recommended outcomes for cSVD trials were stroke occurrence, cognitive decline, dependency, mobility problems, and mood disorders [5], but not neuroimaging. Furthermore, a large percentage of cSVD trials recruited patients with a ‘neuroimaging’ cSVD subtype—this refers to patients with neuroimaging features of cSVD with or without symptoms, but may not have lacunar stroke or cognitive impairment. These trials may have successfully reduced neuroimaging features such as WMH volume in mildly symptomatic or asymptomatic patients. While this demonstrates possible slowing of cSVD development, this does not accurately translate to clinical improvements in patients with severe impairments. While trials that use clinical endpoints may require larger sample sizes to demonstrate a treatment effect, such trials can be less logistically challenging than those using surrogates, as Blair et al. (2024) demonstrate; clinical endpoint trials in VCI can be conducted in less advanced centres, have higher follow-up rates, and more participants randomised [10,31].

Among clinical outcomes used, there was a notable underuse of functional outcomes in cSVD and VCI trials. This was surprising for cSVD given that they are a core recovery endpoint in stroke trials [32]. Higher cSVD burden is also correlated with poorer functional outcomes in acute stroke patients [33]. For VCI, a previous review of cognitive impairment trials found that only 32% of dementia trials and 16% of MCI trials used functional outcomes [34]. Clinical outcomes appeared to heavily focus on cognitive assessments, which are important, but do not capture the holistic effect of the disease. It should be remembered that the defining feature for transition from MCI to dementia is function and not cognition. An emphasis on cognition outcomes over functional outcomes was also reported in other reviews of the field [11,35]. The US Food and Drug Administration (FDA) requires dementia interventions to improve functional ability, not just cognitive assessment scores [10]. Given the importance of independence and quality of life to people living with these conditions [36], functional outcomes should be considered as a key primary outcome for future cSVD and VCI trials.

### 4.4. Strengths and Limitations

The main strength of this study was the comprehensive search strategy. We developed a search syntax for both cSVD and VCI that included various subtypes, definitions, and pathophysiological mechanisms. We also had broad eligibility criteria to include a wide range of studies. We included trials of varying completion status, providing a more accurate overview of current research interests.

However, there were several limitations. In general, cSVD and VCI were poorly indexed in trial registers, leading us to develop a bespoke and at times subjective categorization system for cSVD and VCI subtypes. While we used multiple databases to search the literature, we did not include non-English or specialist registers, and so while our included trials will be broadly representative, we cannot claim to be completely comprehensive in our coverage of the field. Additionally, our study looked at broad categories of selection criteria, interventions, and outcomes, but did not investigate inter-category variability (such as specific cognitive assessments), as this would significantly increase the scale of the review.

### 4.5. Implications for Future Research

Our review demonstrates heterogeneity in all aspects of design, conduct, and reporting of cSVD and VCI intervention trials. The differing interventions used when comparing cSVD and VCI suggest that the two syndromes are being considered in isolation, rather than looking for interventions that may address common underlying mechanisms. The variation in outcomes has been noted in previous reviews [32]. However, the relatively infrequent use of functional and quality of life outcomes is a novel finding, and highlights that trials may not be including those outcomes most important to people living with cSVD and VCI [37]. Robust functional and health-related quality of life measures, suitable for trials, are available [37,38], and we would commend greater use of these assessments in future research.

Variability in the included participants, partly driven by differing inclusion and exclusion criteria, points to the broader issues around the lack of consensus definitions in the cSVD and VCI fields. The criteria used for trial selection also risk creating a non-generalizable evidence base. A particular concern is where those factors that are important risk factors for cSVD or VCI, such as increasing age, comorbidity, or previous stroke, are used as exclusions, and so the included participants are not representative of real-world patients. While best practice statements around improving trial diversity are available, stronger action may be needed to challenge and change current practice. One example is the recent ‘statement of intent’ by the UK Chief Medical Officer, backed by all leading research funders, to stop the arbitrary exclusion of people from trials on the basis of chronological age alone. Professional societies and other interest holder groups in the fields of stroke and dementia could issue similar statements to raise awareness and hopefully improve practice.

## 5. Conclusions

This review highlights substantial variability in all aspects of cSVD and VCI trial design. Generalisability of findings is limited by inconsistent neuroimaging criteria and exclusion of multimorbid patients. Additionally, reliance on surrogate endpoints reflects the need for greater focus on clinical outcomes, such as functional ability, which better reflect treatment impact. The variety of interventions studied suggests a need for a more integrated approach, recognising the overlap in pathophysiologies of cSVD and VCI. Collaboration between the cSVD and VCI communities may lead to increased standardisation of trial design, and we would support the development of core outcome sets for different trial phases and population subtypes. Finally, best practice recommendations for trial design and reporting should be followed where available.

Overall, this review emphasises the importance of refining trial methodologies and reducing inconsistencies to accelerate progress in developing treatments for these common and important conditions.

## Figures and Tables

**Figure 1 neurolint-17-00181-f001:**
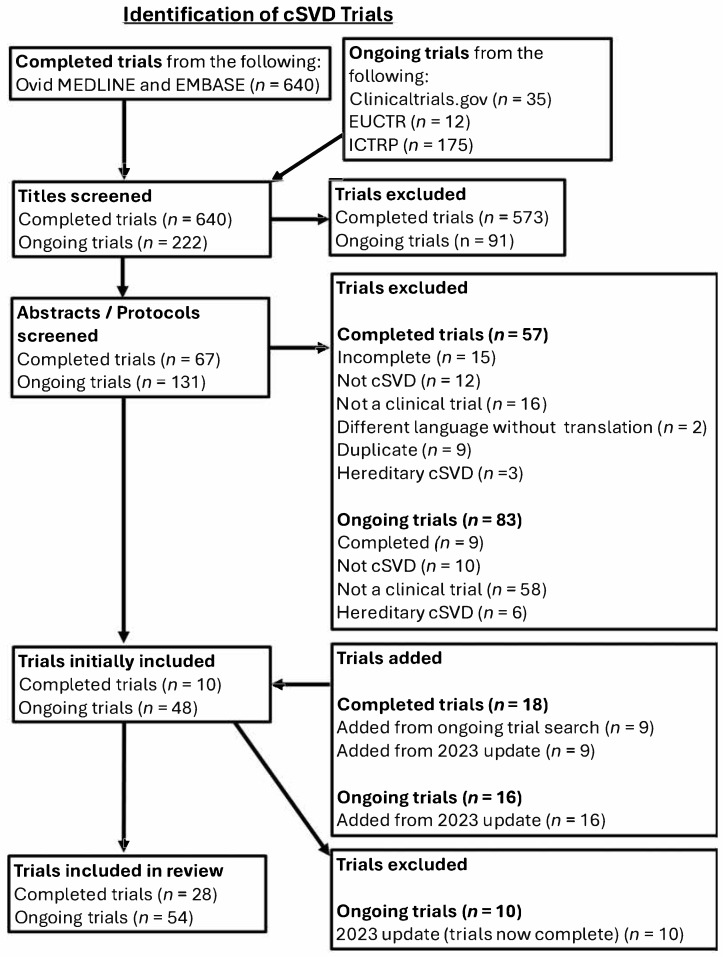
PRISMA chart depicting study selection processes for cSVD trials.

**Figure 2 neurolint-17-00181-f002:**
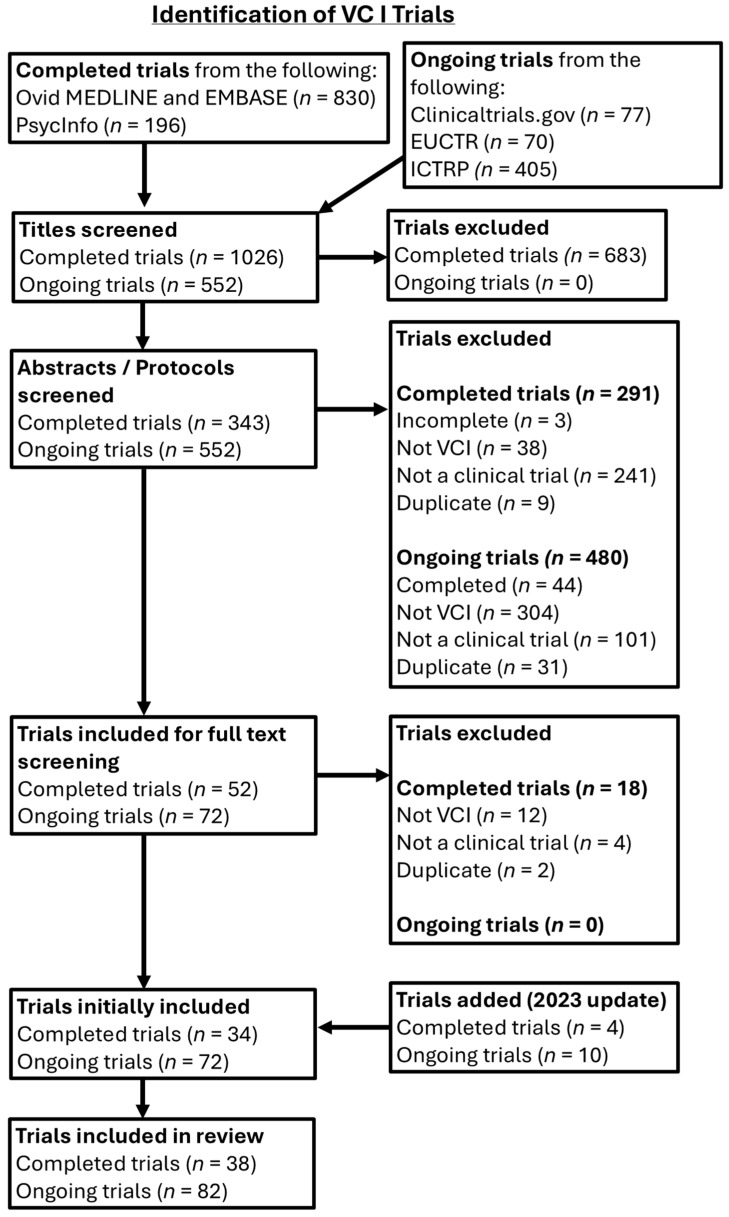
PRISMA chart depicting study selection processes for VCI trials.

**Table 1 neurolint-17-00181-t001:** Categorization system for cSVD and VCI subtypes described in trial populations.

cSVD Subtypes	
Lacunar stroke	Predominantly patients with clinical lacunar stroke
Cognitive impairment	Predominantly patients with cognitive impairment or dementia, and/or whereby a specified cognitive score range is required to enter the trial
Symptomatic neuroimaging	Patients with a range of cSVD neuroimaging features consistent with frank or overt cSVD symptoms, but no history of clinical stroke (e.g., ‘White matter hyperintensities and lacunes consistent with gait impairment’)
Asymptomatic neuroimaging	Patients with a range of cSVD neuroimaging features without frank or overt cSVD symptoms
**VCI Subtypes**	
Stroke-associated VaD	Predominantly patients diagnosed with VaD and a history of stroke
cSVD-associated VaD	Predominantly patients diagnosed with VaD and a history of cSVD or neuroimaging features of cSVD
Undifferentiated VaD	Patients diagnosed with VaD whereby a previous history of stroke/cSVD is not described
Stroke-associated VCI	Predominantly patients diagnosed with VCI and a history of stroke
cSVD-associated VCI	Predominantly patients diagnosed with VCI and a history of cSVD or neuroimaging features of cSVD
Undifferentiated VCI	Patients diagnosed with VCI whereby a previous history of stroke/cSVD is not described
Patients with both VaD and VCI and/or Patients with VaD/VCI risk factors	Trial included both VCI and VaD patients Trial included patients with risk factors for developing VCI or VaD who may not be formally diagnosed with VCI/VaD

cSVD, cerebral small vessel disease. VaD, vascular dementia. VCI, vascular cognitive impairment.

**Table 2 neurolint-17-00181-t002:** Basic Characteristics of cSVD Trials.

Category	Characteristic	Number of cSVD Trials (%)
Trial status	Completed	28 (34.1)
	Ongoing	54 (65.9)
	Active	3 (3.7)
	Completed, not yet published	0 (0.0)
	Not yet recruiting/pending	13 (15.9)
	Recruiting	27 (32.9)
	Suspended/terminated	3 (3.7)
	Unknown	8 (9.8)
Trial phase	Pilot	9 (11.0)
	Phase 1	0 (0.0)
	Phase 2	15 (18.3)
	Phase 3	5 (6.1)
	Phase 4	16 (19.5)
	Unknown/not stated	37 (45.1)
Country/region	North America	9 (11.0)
	Europe and the UK	18 (22.0)
	Asia	52 (63.4)
	Others	3 (3.7)
cSVD subtype	Lacunar stroke	24 (29.3)
	Cognitive impairment	11 (13.4)
	Symptomatic neuroimaging	32 (39.0)
	Asymptomatic neuroimaging	15 (18.3)
Number of participants	Mean number of participants	369
	Standard deviation	1675
Minimum age of eligibility	Mean minimum age of eligibility (years)	41
	Median minimum age of eligibility (years)	45
	Standard deviation	15

cSVD, cerebral small vessel disease. UK, United Kingdom.

**Table 3 neurolint-17-00181-t003:** Basic Characteristics of VCI Trials.

Category	Characteristic	Number of cSVD Trials (%)
Trial status	Completed	38 (31.7)
	Ongoing	82 (68.3)
	Active	3 (3.7)
	Completed, not yet published	0 (0.0)
	Not yet recruiting/pending	20 (16.7)
	Recruiting	50 (41.7)
	Suspended/terminated	1 (0.8)
	Unknown	8 (6.7)
Trial phase	Pilot	13 (10.8)
	Phase 1	5 (4.2)
	Phase 2	17 (14.2)
	Phase 3	14 (11.7)
	Phase 4	14 (11.7)
	Unknown/not stated	57 (47.5)
Country/region	North America	14 (11.7)
	Europe and the UK	13 (10.8)
	Asia	86 (71.7)
	Others	7 (5.8)
VCI subtype	Stroke-associated VaD	4 (3.3)
	cSVD-associated VaD	6 (5.0)
	Undifferentiated VaD	30 (25.0)
	Stroke-associated VCI	29 (24.7)
	CSVD-associated VCI	24 (20.0)
	Undifferentiated VCI	23 (19.2)
	Patients with both VaD and VCI and/or patients with VaD/VCI risk factors	4 (3.3)
Number of participants	Mean number of participants	298
	Standard deviation	1161
Minimum age of eligibility	Mean minimum age of eligibility (years)	41
	Median minimum age of eligibility (years)	45
	Standard deviation	14

cSVD, cerebral small vessel disease. UK, United Kingdom. VaD, vascular dementia. VCI, vascular cognitive impairment.

**Table 4 neurolint-17-00181-t004:** Counts and Percentages of Inclusion and Exclusion Criteria categories used in cSVD Trials.

Selection Criteria Category	Inclusion Criteria (%)	Exclusion Criteria (%)
Comorbidities	0 (0.0)	75 (91.5)
Cognition	34 (41.5)	37 (45.1)
Functional status	17 (20.7)	10 (12.2)
Neuroimaging	72 (87.8)	21 (25.6)
Neurological impairments	25 (30.5)	12 (14.6)
Neuropsychiatric and mood disorders	3 (3.7)	32 (39.0)
Physiological measurements	14 (17.1)	20 (24.4)
Stroke occurrence	32 (39.0)	31 (37.8)
Tissue/Blood/CSF biomarkers	2 (2.4)	7 (8.5)
Vascular risk factors	12 (14.6)	61 (74.4)
Other selection criteria	7 (8.5)	7 (8.5)

CSF, cerebrospinal fluid.

**Table 5 neurolint-17-00181-t005:** Counts and Percentages of Inclusion and Exclusion Criteria categories used in VCI Trials.

Selection Criteria Category	Inclusion Criteria (%)	Exclusion Criteria (%)
Comorbidities	1 (0.8)	107 (89.2)
Cognition	106 (88.3)	42 (35.0)
Functional status	29 (24.2)	3 (2.5)
Neuroimaging	61 (50.8)	7 (5.8)
Neurological impairments	18 (15.0)	24 (20.0)
Neuropsychiatric and mood disorders	7 (5.8)	86 (71.6)
Physiological measurements	7 (5.8)	13 (10.8)
Stroke occurrence	54 (45.0)	21 (17.5)
Tissue/Blood/CSF biomarkers	1 (0.8)	5 (4.2)
Vascular risk factors	24 (20.0)	38 (31.6)
Hachinski Ischaemic Scale	27 (22.5)	0 (0.0)
Other selection criteria	17 (14.2)	4 (3.0)

CSF, cerebrospinal fluid.

**Table 6 neurolint-17-00181-t006:** Counts and Percentages of Intervention Classes used in cSVD and VCI Trials.

Intervention Class	Number of cSVD Trials (%)	Number of VCI Trials (%)
Antihypertensives	11 (13.4)	3 (2.5)
Antithrombotics	4 (4.9)	1 (0.8)
Cognitive enhancers/anti-dementia drugs	4 (4.9)	13 (10.8)
Cognitive rehabilitation	1 (1.2)	21 (17.5)
Exercise and lifestyle interventions	5 (6.1)	22 (18.3)
Physiological interventions	23 (28.0)	13 (10.8)
Traditional Chinese Medicine	10 (12.2)	42 (35.0)
Vasoactive drugs	18 (22.0)	12 (10.0)
Other pharmacological interventions	9 (11.0)	12 (10.0)
Other non-pharmacological interventions	0 (0.0)	1 (0.8)

cSVD, cerebral small vessel disease. VCI, vascular cognitive impairment.

**Table 7 neurolint-17-00181-t007:** Counts and Percentages of Primary and Secondary Outcomes used in cSVD Trials.

Outcome Category	Primary Outcomes (%)	Secondary Outcomes (%)
Clinician assessment	2 (2.4)	2 (2.4)
Cognition	30 (36.6)	32 (39.0)
Economic outcomes	0 (0.0)	0 (0.0)
Functional outcomes	11 (13.4)	19 (23.2)
Neuroimaging (other)	18 (22.0)	19 (23.2)
Neuroimaging (structural)	25 (30.5)	23 (28.0)
Neurological impairments	10 (12.2)	9 (11.0)
Neuropsychiatric and mood disorders	6 (7.3)	18 (22.0)
Physiological measurements	18 (22.0)	30 (36.6)
Safety and adherence outcomes	9 (11.0)	27 (32.9)
Stroke occurrence	3 (3.7)	13 (15.8)
Tissue/blood/CSF biomarkers	3 (3.7)	23 (28.0)
Other outcomes	1 (1.2)	11 (13.4)

CSF, cerebrospinal fluid.

**Table 8 neurolint-17-00181-t008:** Cognitive Assessment Tools used as Primary Outcomes in cSVD Trials.

Cognitive Assessment Type	Cognitive Assessment Tool	Number of Trials Reporting Tool
Global Rating Scales	Addenbrooke’s Cognitive Examination (ACE-III)	1
	Alzheimer’s Disease Assessment Scale-Cognitive Subscale (ADAS-Cog)	4
	Clinician’s Interview-Based Impression of Change Plus Caregiver Input (CIBIC-plus)	2
	Hasegawa Dementia Scale	1
	Mini-Mental State Examination (MMSE)	8
	Montreal Cognitive Assessment (MoCA)	12
	Toronto Cognitive Assessment (TorCA)	1
	Vascular Dementia Assessment Scale-Cognitive Subscale (VADAS-Cog)	1
Single Domain Assessments	Auditory Verbal Learning Test	2
	Boston Naming Test	1
	Brief Visuospatial Memory Test-Revised	1
	Controlled Oral Word Association Test	1
	Hopkin’s Verbal Learning Test	1
	Memory and Executive Screening Test (MES)	1
	Judgement of Line Orientation Test	1
	Symbol Digits Modalities Test	1
	Stroop Colour and Word Test	1
	Trail Making Test A or B (or Colour Trails Test)	2
	Verbal Fluency Test	1
Neuropsychological Batteries	Frontal Assessment Battery (FAS)	1
	National Institute of Neurological Disorders and Stroke-Canadian Stroke Network (NINDS-CSN) Protocol	1
Unspecified Cognitive Assessments		8

**Table 9 neurolint-17-00181-t009:** Functional Assessment Tools used as Primary Outcomes in cSVD Trials.

Functional Assessment Tool	Number of Trials Reporting Tool
Barthel Index of Activities of Daily Living	1
Lawton’s Instrumental Activities of Daily Living (IADL)	1
Unspecified ‘ADL’ Assessment Tool	9

ADL, Activities of Daily Living.

**Table 10 neurolint-17-00181-t010:** Counts and Percentages of Primary and Secondary Outcomes used in VCI Trials.

Outcome Category	Primary Outcomes (%)	Secondary Outcomes (%)
Clinician assessment	8 (6.0)	10 (8.3)
Cognition	104 (86.0)	64 (53.0)
Economic outcomes	0 (0.0)	2 (1.6)
Functional outcomes	15 (12.5)	50 (41.6)
Neuroimaging (other)	9 (7.5)	16 (13.0)
Neuroimaging (structural)	7 (5.8)	17 (14.2)
Neurological impairments	4 (3.0)	16 (13.0)
Neuropsychiatric and mood disorders	11 (9.2)	34 (28.3)
Physiological measurements	9 (7.5)	19 (15.8)
Safety and adherence outcomes	4 (3.0)	13 (10.8)
Stroke occurrence	2 (1.6)	5 (4.2)
Tissue/blood/CSF biomarkers	12 (10.0)	22 (18.3)
Other outcomes	9 (7.5)	15 (12.5)

CSF, cerebrospinal fluid.

**Table 11 neurolint-17-00181-t011:** Cognitive Assessment Tools used as Primary Outcomes in VCI Trials.

Cognitive Assessment Type	Cognitive Assessment Tool	Number of Trials Reporting Tool
Global Rating Scales	Alzheimer’s Disease Assessment Scale-Cognitive Subscale (ADAS-Cog)	24
	Alzheimer’s Disease Cooperative Study—Clinicians’ Global Impression of Change (ADCS-CGIC)	3
	Brief Cognitive Assessment Tool (BCAT)	2
	Clinical Dementia Rating (CDR)	5
	Clinical Global Impressions Scale (CGI)	1
	Clinician’s Interview-Based Impression of Change Plus Caregiver Input (CIBIC-plus)	4
	Clock Drawing Test	7
	Cognitive Change Index	1
	Cognivue Scale	1
	Hasegawa Dementia Scale	3
	Mattis Dementia Rating Scale (MDRS)	1
	Mini-Mental State Examination (MMSE)	29
	Montreal Cognitive Assessment (MoCA)	31
	Quick Dementia Rating System (QDRS)	1
	Vascular Dementia Assessment Scale-Cognitive Subscale (VADAS-Cog)	12
Single Domain Assessments	Auditory Verbal Learning Test	4
	Brief Visuospatial Memory Test-Revised	2
	California Verbal Learning Test	1
	Controlled Oral Word Association Test	4
	Deary–Liewald Reaction Time Task	1
	Delis–Kaplan Executive Function System	1
	Digit Span Test	5
	Digit Symbol Substitution Test	3
	Eriksen Flanker Task	1
	Executive Interview (EXIT-25)	1
	Hopkin’s Verbal Learning Test	4
	Judgement of Line Orientation Test	1
	Keep Track Test	1
	Mental Rotation Test	1
	Mnemonic Similarity Task	1
	North American Reading Test	1
	Number Letter Task	1
	Phonemic Fluency Test	1
	Rey–Osterrieth Complex Figure Test	2
	Semantic Fluency Test	1
	Symbol Digits Modalities Test	7
	Stroop Colour and Word Test	9
	Trail Making Test A or B (or Colour Trails Test)	16
	Verbal Fluency Test	7
	Verbal Paired Associates Task	1
	Wisconsin Card Sorting Test	1
Neuropsychological Batteries	Frontal Assessment Battery (FAS)	4
	Loewenstein Occupational Therapy Cognitive Assessment (LOTCA)	3
	National Institute of Neurological Disorders and Stroke-Canadian Stroke Network (NINDS-CSN) Protocol	2
	NeuroCart Battery	1
	National Institute of Health (NIH) Toolbox Cognitive Battery	1
Unspecified Cognitive Assessments		21

**Table 12 neurolint-17-00181-t012:** Functional Assessment Tools used as Primary Outcomes in VCI Trials.

Functional Assessment Tool	Number of Trials Reporting Tool
Alzheimer’s Disease Cooperative Study Activities of Daily Living Inventory (ADCS-ADL)	2
Disability Assessment for Dementia (DAD)	2
EuroQoL Instruments	1
Instrumental Activities of Daily Living (IADL)	2
Lawton’s Instrumental Activities of Daily Living (IADL)	1
Short Form Health Survey (SF-36)	1
Standard Assessment of Global Everyday Activities (SAGEA)	1
Unspecified ‘ADL’ Assessment Tool	6
World Health Organization Quality of Life Scale (WHOQOL-BREF)	1

ADL, Activities of Daily Living.

## Data Availability

The original contributions presented in this study are included in the article/Appendix A. Further inquiries can be directed to the corresponding author.

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
