# Peer review of "Design of Trials for Cerebral Small Vessel Disease and Vascular Cognitive Impairment"

_2035-8377, 2025, doi:10.3390/neurolint17110181_

Round 1
Reviewer 1 Report
Comments and Suggestions for Authors
The manuscript entitled “Design of Trials for Cerebral Small Vessel Disease and Vascu- 2 lar Cognitive Impairment” presents a timely and comprehensive review of clinical trials in cerebral small vessel disease (cSVD) and vascular cognitive impairment (VCI). The topic is highly relevant given the growing recognition of vascular contributions to cognitive decline. The authors have compiled a substantial dataset and provided valuable insights into trial design, outcome measures, and participant selection. However, several areas require clarification and improvement to enhance the manuscript’s clarity, consistency, and impact. There are as follows:
- Include a table summarizing cognitive and functional assessment tools used across trials.
- Authors need to discuss strategies to improve inclusivity in future trials.
- The figures in the manuscript are informative but lack visual clarity, which may hinder interpretation. I recommend enhancing the resolution and overall quality of the figures to ensure that labels, legends, and data points are easily readable.
- The manuscript notes the diversity of outcome measures, but it does not analyze which tools are most used or effective. It would be good to include a table summarizing cognitive and functional assessment tools used across trials.
- Please ensure consistent use of terminology throughout the manuscript. For example, in Figure 2, the label refers to “VCI/VaD,” while the legend only mentions “VCI.” This inconsistency may confuse readers, especially those unfamiliar with the distinctions between vascular cognitive impairment (VCI), vascular dementia (VaD), and vascular mild cognitive impairment (vascular MCI). I recommend standardizing the terminology across figures, legends, and text to improve clarity and coherence.
Author Response
Thank you very much for taking the time to review this manuscript and for your helpful and insightful comments. Please see our responses to your comments below in bold. Our revised manuscript contains new and modified text which are highlighted in yellow. In addition, we have recategorized our paper as a scoping review. This was felt to be a more appropriate definition for our review rather than a systematic review as we provide a broad descriptive synthesis of the trial design and methods in cerebral small vessel disease (cSVD) and vascular cognitive impairment (VCI). This has been discussed and agreed with the editors.
The manuscript entitled “Design of Trials for Cerebral Small Vessel Disease and Vascular Cognitive Impairment” presents a timely and comprehensive review of clinical trials in cerebral small vessel disease (cSVD) and vascular cognitive impairment (VCI). The topic is highly relevant given the growing recognition of vascular contributions to cognitive decline. The authors have compiled a substantial dataset and provided valuable insights into trial design, outcome measures, and participant selection.
Response: We thank the reviewer for commending the relevance of the topic and noting the work required to create the dataset on which the analysis is based.
However, several areas require clarification and improvement to enhance the manuscript’s clarity, consistency, and impact. There are as follows:
Include a table summarizing cognitive and functional assessment tools used across trials.
Response: Thank you for the suggestion. We have created new tables summarising cognitive and functional assessment tools used as primary outcomes across cSVD and VCI trials, while also quantifying the number of trials that used each tool as part of their primary outcomes. Tables 8 and 9 summarise cognitive and functional assessment tools used in cSVD trials (page 15). Tables 11 and 12 summarise cognitive and functional assessment tools used in VCI trials (pages 17-18).
Authors need to discuss strategies to improve inclusivity in future trials.
Response: We agree that inclusivity is a major issue for trials. Indeed, we have been involved in creating best practice materials to improve inclusivity. We have added text to the Discussion, outlining the issues and suggesting strategies to improve inclusion.
The figures in the manuscript are informative but lack visual clarity, which may hinder interpretation. I recommend enhancing the resolution and overall quality of the figures to ensure that labels, legends, and data points are easily readable.
Response: Thank you for pointing this out. We have created new PRISMA charts with improved resolution and larger text. These can be found on page 8 (for cSVD trials) and page 9 (for VCI trials).
The manuscript notes the diversity of outcome measures, but it does not analyze which tools are most used or effective. It would be good to include a table summarizing cognitive and functional assessment tools used across trials.
Response: We agree and have created new tables summarising the cognitive and functional outcomes.
Please ensure consistent use of terminology throughout the manuscript. For example, in Figure 2, the label refers to “VCI/VaD,” while the legend only mentions “VCI.” This inconsistency may confuse readers, especially those unfamiliar with the distinctions between vascular cognitive impairment (VCI), vascular dementia (VaD), and vascular mild cognitive impairment (vascular MCI). I recommend standardizing the terminology across figures, legends, and text to improve clarity and coherence.
Response: Thank you for pointing out this inconsistency. We have changed all titles and labels relating to data from VCI trials to only include the term ‘VCI’. However, we still reference the term ‘VaD’ in some parts of the text in Methods and Discussion to refer to the most severe presentation of VCI. We clarify in the introduction that the term ‘VCI’ represents a continuum of cognitive impairment, ranging from mild (vascular MCI) to severe (VaD) (line 65). This should hopefully make it clearer to readers that VaD is not a distinct disease but rather within the spectrum of VCI.
Reviewer 2 Report
Comments and Suggestions for Authors
This is a systematic review evaluating the design of clinical trials for cerebral small vessel disease and vascular cognitive impairment. The authors analyzed 82 CBMs studies and 120 SCN studies. They report significant variability in approaches to population selection, choice of interventions, and endpoints. The authors note that functional outcomes have been underused in studies of both categories. As for the research issue, it is relevant because the non-standardized design of research limits the generalizability of results and slows down the development of treatment methods. The manuscript has a clear structure and is easy to read. The authors need to consider the following issues:
1) According to the recommendations of PRISMA, it is necessary to indicate whether the study has been registered. The authors could clarify this aspect.
2) The authors reasonably point out the imperfection of modern categorical classifications, while offering their own. Why was it decided to select just such categories in this article? (Table 1. Categorization system for cSVD and VCI subtypes described in trial populations)
3) Correcting some design issues may bring the manuscript to a more finished look. In the second illustration (Figure 2. PRISMA chart depicting study selection processes for VCI trials), the word VaD is underlined in red at the top. Page 8 of the manuscript is almost empty.
4) The authors point out the limitations imposed by the age criteria for inclusion/exclusion. How appropriate is it to use the term "ageism" in a scientific article in the context of criticism of these criteria? Perhaps a more neutral term would be appropriate to describe the current situation. (line 370)
Author Response
Thank you very much for taking the time to review this manuscript and for your helpful and insightful comments. Please see our responses to your comments below in bold. Our revised manuscript contains new and modified text which are highlighted in yellow. In addition, we have recategorized our paper as a scoping review. This was felt to be a more appropriate definition for our review rather than a systematic review as we provide a broad descriptive synthesis of the trial design and methods in cerebral small vessel disease (cSVD) and vascular cognitive impairment (VCI). This has been discussed and agreed with the editors.
This is a systematic review evaluating the design of clinical trials for cerebral small vessel disease and vascular cognitive impairment. The authors analyzed 82 CBMs studies and 120 SCN studies. They report significant variability in approaches to population selection, choice of interventions, and endpoints. The authors note that functional outcomes have been underused in studies of both categories. As for the research issue, it is relevant because the non-standardized design of research limits the generalizability of results and slows down the development of treatment methods. The manuscript has a clear structure and is easy to read.
Response: We thank the reviewer for encouraging words around the structure and text.
The authors need to consider the following issues:
- According to the recommendations of PRISMA, it is necessary to indicate whether the study has been registered. The authors could clarify this aspect.
Response: Thank you for this suggestion. The review is registered on the International Platform of Registered Systematic Review and Meta-analysis Protocols resource (INPLASY). We have now provided the registration number and hyperlink as part of the Methods section (page 3, line 137).
- The authors reasonably point out the imperfection of modern categorical classifications, while offering their own. Why was it decided to select just such categories in this article? (Table 1. Categorization system for cSVD and VCI subtypes described in trial populations)
Response: We recognise that there is a lack of consensus and consistency in the definitions of cSVD and VCI. This became increasingly apparent during the data extraction process, as noted by the substantial variability in patient populations for both conditions. It was not our intention to add to the heterogeneity by creating new definitions for others to use. Rather, we created in-house definitions to be operational and used purely for the datasets in this review, as an attempt to convey the type of cSVD/VCI patient population that these trials were studying.
- Correcting some design issues may bring the manuscript to a more finished look. In the second illustration (Figure 2. PRISMA chart depicting study selection processes for VCI trials), the word VaD is underlined in red at the top. Page 8 of the manuscript is almost empty.
Response: Thank you for pointing this out. We have created new PRISMA charts with improved resolution and larger text. These can be found on page 8 (for cSVD trials) and page 9 (for VCI trials). We have also standardized the terminology, using the term ‘VCI’ when referring to data relating to VCI trials.
- The authors point out the limitations imposed by the age criteria for inclusion/exclusion. How appropriate is it to use the term "ageism" in a scientific article in the context of criticism of these criteria? Perhaps a more neutral term would be appropriate to describe the current situation. (line 370)
Response: Age is one of the protected characteristics, and deliberate exclusion of someone from participating in a trial on the basis of age would meet the legal definition of ageism. This issue is gaining increasing attention, for example in the UK, the Chief Medical Officer, backed by all major research funders, has issued a statement of intent to halt ageism in clinical research. We mention this is the Discussion section.
Round 2
Reviewer 1 Report
Comments and Suggestions for Authors
NA